# Preservation of Mitochondrial Function by SkQ1 in Skin Fibroblasts Derived from Patients with Leber’s Hereditary Optic Neuropathy Is Associated with the PINK1/PRKN-Mediated Mitophagy

**DOI:** 10.3390/biomedicines12092020

**Published:** 2024-09-04

**Authors:** Jin Xu, Yan Li, Shun Yao, Xiuxiu Jin, Mingzhu Yang, Qingge Guo, Ruiqi Qiu, Bo Lei

**Affiliations:** 1Zhengzhou University People’s Hospital, Henan Provincial People’s Hospital, Zhengzhou 450003, China; 2Academy of Medical Sciences, Zhengzhou University, Zhengzhou 450052, China; 3Henan Eye Institute, Henan Eye Hospital, Zhengzhou University People’s Hospital, Henan Provincial People’s Hospital, Zhengzhou 450003, China; 4Eye Institute, Henan Academy of Innovations in Medical Science, Zhengzhou 451163, China

**Keywords:** ROS, mitochondria function, SkQ1, mitophagy, Leber’s hereditary optic neuropathy

## Abstract

Increased or altered mitochondrial ROS production in the retinal ganglion cells is regarded as the chief culprit of the disease-causing Leber’s hereditary optic neuropathy (LHON). SkQ1 is a rechargeable mitochondria-targeted antioxidant with high specificity and efficiency. SkQ1 has already been used to treat LHON patients, and a phase 2a randomized clinical trial of SkQ1 has demonstrated improvements in eyesight. However, the underlying mechanism of SkQ1 in LHON remains unclear. This study aimed to assess the effects and molecular mechanism of SkQ1 in the preservation of mitochondrial function using skin fibroblasts derived from LHON patients. Our study found that SkQ1 could reduce ROS production and stabilize the mitochondrial membrane. Mechanistically, through network pharmacology and molecular docking, we identified the key targets of SkQ1 as SOD2 and PINK1, which play crucial roles in redox and mitophagy. SkQ1 interacted with PINK1 and downregulated its expression to balance mitochondrial homeostasis. Collectively, the findings of our study reveal that by regulating PINK1/PRKN-mediated mitophagy, SkQ1 preserves mitochondrial function in LHON fibroblasts. The data indicate that SkQ1 may be a novel therapeutic intervention to prevent the progression of LHON.

## 1. Introduction

With penetrance rates of up to 50–60% in males and 10–20% in females, Leber’s hereditary optic neuropathy (LHON) is a mitochondrial disorder that primarily affects young males [1]. Most classic LHON cases in populations result from one of three mtDNA mutation genes that encode subunits of respiratory chain complex I: G11778A, T14484C, and G3460A [2]. These mutations induce mitochondrial dysfunction within retinal ganglion cells (RGCs), ultimately causing optic nerve atrophy and the onset of acute or subacute bilateral vision impairment, concurrently or consecutively [3]. The pathogenesis of LHON is not yet fully understood, but one plausible hypothesis is that the imbalance of redox homeostasis renders the RGCs vulnerable to apoptosis [4,5].

Up to now, therapeutic strategies for LHON have mainly focused on antioxidants, but with limited effect [2]. Antioxidants like Idebenone, EPI 743, and Bendavia, which play a role in reducing oxidative stress, have been tested in clinical trials [6,7,8]. However, these new medicines have shown limited efficacy in scavenging ROS [9]. Gene therapy medicinal products are also being explored to improve the visual prognosis in LHON [10]. However, the efficacy of AAV transfection in gene therapy needs further enhancement, which necessitates addressing the abnormal mitochondrial membrane potential, potentially caused by elevated levels of ROS [8]. Thus, there is a critical need to find novel drugs with high efficiency and low cytotoxicity for scavenging ROS.

SkQ1 is a rechargeable, mitochondria-targeted antioxidant with high specificity, efficiency, and security [11]. This lipophilic antioxidant specifically targets the mitochondrial matrix [12]. The SkQ1 concentration in the mitochondrial matrix could be 1 × 10^8^ times higher than in the extracellular space under certain conditions [13]. SkQ1 drops have demonstrated statistically significant improvement of BCVA relative to baseline in LHON patients [14] and the potential to improve uveitis, wet age-related macular degeneration (AMD), and neurodegenerative, cardiovascular, and metabolic disorders during the early stages of clinical development in previous studies [15]. Additionally, a clinical study (NCT02121301) demonstrated the safety and efficacy of SkQ1 in treating dry eye disease [16]. SkQ1 may target and regulate mitochondria-associated proteins to protect and improve mitochondrial function. In this study, we intended to observe the effect of SkQ1 in patient fibroblasts carrying the m.G11778A mutation and to explore the underlying molecular mechanism.

## 2. Materials and Methods

### 2.1. Cells Construction and Culturing

Human skin fibroblasts were derived from one healthy volunteer and two LHON patients in different families. The cells were isolated from 2 mm^2^ dermal punch biopsies and were transfected with the SV40 virus to construct immortalized fibroblast cell lines (IFBs). The IFBs were generated with informed consent and ethics approval (HNEECKY-2019(12)). The IFBs were cultured in Dulbecco’s Modified Eagle’s Medium (DMEM)/Hams F-12 50/50 Mix (10-092-CV, Corning, New York, NY, USA) supplemented with 10% FBS (35-030-CV, Corning, New York, NY, USA) and 10 ng/mL basic fibroblast growth factor (Labsystech, Shanghai, China). HEK293T cells were maintained in DMEM high-glucose medium (10-013-CV, Corning, New York, NY, USA) containing 10% fetal bovine serum (35-030-CV, Corning, New York, NY, USA) and 1% Penicillin-Streptomycin Solution 100× (P1400, Solarbio, Beijing, China). RPE-1 cells were cultured in Dulbecco’s Modified Eagle’s Medium (DMEM)/Hams F-12 50/50 Mix (10-092-CV, Corning, New York, NY, USA) supplemented with 10% fetal bovine serum. The above three types of cells were cultured at 37 °C in a 5% CO_2_ thermostatic incubator. HEK293T and HTERT RPE-1 cell lines were purchased from the American Type Culture Collection (ATCC, Manassas, VA, USA).

### 2.2. SkQ1 Stock and Working Solutions

The stock solution of SkQ1 (HY-100474, MCE, Shanghai, China) was dissolved in anhydrous dimethyl sulfoxide (DMSO) solution. The SkQ1 stock concentration was 10 mmol/L. Following the instructions, this stock was stored at −80 °C until use, and it can be stored for 6 months from the preparation date. The working solutions of SkQ1 were prepared as 100 × dilutions in the maintenance medium as described above. Both the stock solution and working dilutions were kept in light-proof containers and away from moisture before and after being applied to the cultures.

### 2.3. Cell Viability Detection

#### 2.3.1. Cell Viability, Dosing, and Safety Ranges

LHON-IFBs were cultured overnight in a 96-well plate. The next day, the IFBs were treated with different concentrations from 2 nM to 20,000 nM to evaluate the proper dosing and safety range of SkQ1. In the meantime, all complete culture media with or without SkQ1 were changed once every 2 days. Additionally, the quantitative cell viability was determined by the CCK8 assay (CK04, DOJINDO, Kyushu, Japan). The CCK8 was diluted with basal medium without serum at a ratio of 1:10 to prepare the CCK8 working solution. The CCK8 working solution (100 µL/well) was added, and the IFBs were incubated for 4 h. The absorbance was detected at 450 nm. The value at 0 h was taken as the origin to calculate the relative value at each time point.

#### 2.3.2. Cell Viability with Long-Term SkQ1 Treatment

The cells were cultured overnight in a 96-well plate. The next day, the cells were treated with 20 nM SkQ1. All complete culture media with or without SkQ1 were changed once every 2 days during the next couple of days. The CCK8 determination method was the same as described above.

#### 2.3.3. Cell Viability with H_2_O_2_ Treatment

The cells were cultured overnight in a 96-well plate. The next day, the cells were pretreated with 20 nM SkQ1 for 2.5 h. A 200 μmol/L volume of H_2_O_2_ was added to treat the cells for 4 h. The CCK8 assay determined the cell viability.

### 2.4. Intramitochondrial Superoxide Detection

LHON-IFBs were pretreated with 20 nM SkQ1 or vehicle (DMSO) for 7 days, then both negative control (NC) and LHON-IFBs were seeded in 35 mm dishes cultured overnight, then treated with 200 μM H_2_O_2_ for 45 min. The mitochondrial superoxide was measured by MitoSOX^TM^ red reagent (M36008, Invitrogen, California, CA, USA). Cells were incubated with 1 µM MitoSOX^TM^ red reagent working solution diluted with HBSS (BL561A, Biosharp, Heifei, China) for 30 min at 37 °C. Then, the cells were washed gently three times with warm HBSS. Fluorescent images were visualized with single-photon confocal microscopy equipped with a 40× oil objective (excitation 396 nm, emission 610 nm) with fixed scanning parameters (Gain: 2×, exposure time: 120 ms). Unstained cells were the background. The mean intensity of MitoSOX fluorescence was analyzed using ImageJ software. Calibrated Cumulative Optical Density = Cumulative Optical Density − Cell Area to be Measured × Average Background Optical Density. We used the fluorescence values of the negative control NC group as an internal reference and homogenized all data to exclude differences due to dye uptake.

### 2.5. Mitochondrial Function Assay

#### 2.5.1. Mitochondrial Membrane Potential Assay

The mitochondrial membrane potential (MMP) assay was performed using a JC-1 detection kit (C2006, Beyotime, Shanghai, China). The cells were incubated overnight in a 6-well plate. After washing the cells once with PBS, 1 mL JC-1 staining working solution and 1 mL serum-containing medium were added to each well. Cells were cultured at 37 °C in an incubator for 20 min. After incubation, the supernatant was discarded, and the cells were washed twice with JC-1 buffer. Finally, 2 mL of serum-containing medium was added to the cell wells, and fluorescent images were taken. The green-to-red fluorescence intensity ratio represents the mitochondrial membrane potential, which is inversely proportional.

#### 2.5.2. Mitochondrial Permeability Transition Pore Assay

The degree of the mitochondrial permeability transition pore opening was detected by the MPTP assay kit (C2009S, Beyotime), which employed Calcein AM, a fluorescent probe for membrane permeability. The cells were inoculated overnight in 12-well plates. After aspirating the culture medium and the cells and washing the cells twice with PBS, 500 μL fluorescence quenching working solution was added to each well. Cells were incubated away from light at 37 °C for 40 min. After incubation, the supernatant was discarded, and the cells were cultured with fresh pre-warmed culture medium away from light at 37 °C for 30 min. After discarding the culture medium and washing with PBS thrice, 2 mL assay buffer was added to each well. The green fluorescent images were observed with a microscope.

#### 2.5.3. Mitochondrial Stress Test Assay

The cells’ oxygen consumption rate (OCR) was measured by Seahorse XFe96 (Agilent Seahorse Technologies, Palo Alto, CA, USA). Key parameters, including basal respiration, ATP-linked respiration, and maximal respiration, were obtained by sequentially adding drugs targeting the mitochondrial electron transport chain (ETC). The specific steps are as follows. The Seahorse XFe96 assay system was switched on and preheated the day before the experiment. Cells were inoculated into the special 96-well cell culture plates with 1.5 × 10^4^ cells per well and grown overnight at 37 °C in a 5% CO_2_ incubator. The probe plate was aligned with each well of the hydration plate so that the Sensor was submerged in sterile water. On the day of the experiment, the cell culture medium was replaced with XF medium containing 2 mM glutamine, 1 mM pyruvate, and 10 mM glucose, and the cell culture plates were allowed to stand for 1 h at 37 °C in a CO_2_-free incubator to exclude gas interferences. A 15 µM volume of oligomycin, 20 µM FCCP, 5 µM antimycin A, and 5 µM rotenone were injected into the probe plate during the assay. Cell counts were normalized and analyzed using the software Wave 2.6.3.

### 2.6. Network Pharmacology and Molecular Docking

Candidate targets of SkQ1 were screened from STITCH (http://stitch.embl.de/, accessed on 16 April 2022), SEA (similarity ensemble approach, https://sea.bkslab.org/, accessed on 21 April 2022), and TargetNet (http://targetnet.scbdd.com/, accessed on 21 April 2022). LHON-related genes were obtained from the Therapeutic Target Database (https://db.idrblab.net/ttd/, accessed on 21 March 2022), OMIM (https://www.omim.org/, accessed on 26 April 2022), and GeneCards (https://www.genecards.org/, accessed on 25 April 2022). The overlapping proteins were defined as the potential targets of SkQ1. DAVID (https://david.ncifcrf.gov/, accessed on 26 April 2022) and Metascape (https://www.metascape.org, accessed on 25 April 2022) online tools were used to perform gene ontology GO (https://www.geneontology.org/, accessed on 13 May 2022) and Kyoto Encyclopedia of Genes and Genome (KEGG, https://www.genome.jp/kegg/, accessed on 25 April 2022) pathway enrichment analysis. In addition, protein–protein interaction (PPI) network and topological analysis were conducted through the STRING (https://cn.string-db.org/, accessed on 6 April 2022) network platform and Cytoscape 3.9.1 software. Further, the affinity coefficients between SkQ1 and potential targets were calculated by molecular docking [17].

### 2.7. Western Blot

Cells were collected with cell scrapers (CSC011025, JET BIOFIL, Guangzhou, China) into 1.5 mL centrifuge tubes on ice. Next, the samples were centrifuged at 1000× *g* for 3 min at 4 °C. Subsequently, samples were lysed in RIPA lysis buffer (PC101, Epizyme Biomedical Technology, Shanghai, China) with 1 × protease inhibitor cocktail (GRF101, Epizyme Biomedical Technology, Shanghai, China) and 1 × phosphatase inhibitor. Afterward, the samples were incubated on ice for 30 min. After centrifugation at 12,000 rpm for 15 min, the supernatants were mixed using a 5 × loading buffer. Moreover, the supernatants were boiled at 95 °C for 5 min. The amount of protein samples loaded was 25 μg. The samples were electrophoresed using 12.5% SDS-PAGE (PG113, Epizyme Biomedical Technology) and transfected into PVDF membranes (IPVH00010, Millipore, Massachusetts, MA, USA). After blocking with 5% milk powder (PS112L, Epizyme Biomedical Technology) for 1.5 h, the membranes were incubated with primary antibodies overnight at 4 °C: PINK1 (6946S, CST, Massachusetts, MA, USA), PRKN (32833, CST, MA, USA), SOD2 (ET1701-54, HuaBio, Hangzhou, China), VDAC1 (ET1601-20, HuaBio, Hangzhou, China), LC3B (ET1701-65, Hangzhou, China) and β-actin (3700S, CST, MA, USA). After washing thrice, the membranes were incubated with the second antibody-HRP conjugate for 1.5 h at room temperature and visualized with a chemiluminescent detection reagent (WBKLS0500, Millipore, MA, USA). Bands were analyzed using the ImageJ software (Version 1.52a, NIH).

### 2.8. Cellular Thermal Shift Assay

Cells were grown to 80% confluency before being treated with 20 μM SkQ1 (or DMSO) for 6 h. Subsequently, they were harvested by cell scrapers (CSC011025, JET BIOFIL) and centrifuged at 1000× *g* for 3 min at 4 °C. Next, the samples were resuspended in 1× PBS (pH = 7.4) with a 1× protease inhibitor cocktail (GRF101, Epizyme Biomedical Technology). Afterward, the cells were aliquoted into PCR tubes and heated for 3 min. The cells were lysed by 4 cycles of freeze–thawing with liquid nitrogen and centrifugation at 12,000 rpm for 15 min. The soluble fractions were analyzed by SDS–PAGE and immunoblotted with antibodies.

### 2.9. Statistical Analysis

GraphPad Prism 8.0 was used for statistical analyses. ImageJ software (Version 1.52a, NIH) was used to analyze fluorescence intensity and band intensity. Data in the graphs were shown as means ± standard deviations. The unpaired *t*-test was used to evaluate the differences. Differences between multiple groups were analyzed using the one-way ANOVA test, supplemented by Bonferroni calibration. All analyses were based on at least three or four independent experiments. The number of each sample was more than 3 (*n* ≥ 3). The statistical significance was set at *p* ≤ 0.05.

## 3. Results

### 3.1. LHON-IFBs Presented Excessive Superoxide and Mitochondrial Dysfunction

Two probands, LHON-I and LHON-II, were recruited for this study. LHON-I, a 28-year-old male, carried the mutation m.11778G > A, and had a BCVA of 0.25 in the right eye and 0.20 in the left eye. LHON-II, a 31-year-old male, had a BCVA of 0.02 in both eyes and carried mutations m.11778G > A and m.10398A > G. With permission, skin tissues were obtained from the two probands, and immortalized skin fibroblasts were generated [18,19].

Fibroblasts donated by healthy volunteers were used as control (NC). Mitochondrial superoxide level was detected by MitoSOX red reagent. The average fluorescent intensity in the LHON fibroblasts was higher than that in the control (Figure 1A,B). Further, when the mitochondrial membrane potential (MMP) is depolarized, JC1 aggregates are broken down into monomers. The results showed that the ratio of monomers to aggregates was higher in both LHON-I and LHON-II than in the NC, suggesting that the MMP was depolarized in LHON-IFBs (Figure 1C,D). Additionally, depolarization of MMP leads to increased mitochondrial membrane permeability, resulting in quenching of fluorescence within the mitochondria. The MPTP fluorescence images unveiled a reduced intensity of green fluorescence in LHON-IFBs, signifying further the impaired mitochondrial membrane potential (Figure 1E,F). Using the Seahorse cellular energy metabolism analyzer to detect changes in dissolved oxygen in real time via optical sensors in synchrony, the results showed that the mitochondrial oxygen consumption rate (OCR) of LHON-IFBs was lower than that of the control group (Figure 1G,H), which indicated that mitochondrial respiration was weakened in LHON-IFBs. The above results suggested the presence of excess superoxide and mitochondrial dysfunction in LHON-IFBs.

### 3.2. Drug Safety of SkQ1

To determine the appropriate and safe dose, we used the CCK8 kit to evaluate the effect of SkQ1 on cell viability. The graph displayed that there was a gradual rise in the LHON-IFBs with treatment concentrations of SkQ1 ranging from 2 nM to 200 nM, despite the cell viability being inhibited beyond 2 μM SkQ1 (Figure 2A,B). In addition, the pictures showed continual growth in cell viability during the treatment period with 20 nM SkQ1 for 9 days (Figure 2C,D). They revealed no significant differences in cell viability among the three groups, indicating that 20 nM was the available concentration with low cytotoxicity. Moreover, after treatment with H_2_O_2_ for 4 h, it was found that the cell viability of cells pretreated with SkQ1 was significantly higher than that of the control (Figure 2E,F), implying that SkQ1 could prevent apoptosis. We obtained the same results in HEK293T and RPE-1 (Figure 2G,H).

### 3.3. SkQ1 Scavenged Mitochondrial Superoxide

To further validate the effect of SkQ1 in eliminating mitochondrial superoxide, we used the MitoSOX™ Red reagents to detect the superoxide in mitochondria. The results showed that there was a significant reduction in mitochondrial superoxide in LHON-IFBs after SkQ1 (20 nM/7 days) treatment compared with the LHON-DMSO (Figure 3A upper part and Figure 3B left part). After treatment with 200 μM H_2_O_2_, there was a significant increase in mitochondrial superoxide in the LHON-DMSO. In contrast, the group pretreated with 20 nM SkQ1 for 7 days displayed a significant decrease in ROS levels (Figure 3A lower part and Figure 3B right part). The results above further validated that SkQ1 can target and scavenge mitochondrial ROS and be used in the early stages of LHON to prevent exacerbation.

### 3.4. SkQ1 Protected Mitochondrial Function

#### 3.4.1. SkQ1 Promoted Mitochondrial Membrane Potential Stability

Based on the performance of SkQ1 in scavenging ROS, we conducted a JC-1 assay to detect mitochondrial membrane potential (MMP, ∆Ψm). When the MMP decreased, JC-1 changed from aggregates to a monomer form, which could produce green fluorescence, so the relative ratio of red to green fluorescence was used to measure the proportion of mitochondrial depolarization. The results illustrated that there was a slight increase in MMP in LHON-SkQ1 than that in LHON-DMSO (Figure 4A upper part and Figure 4B left part). After treatment with 200 μM H_2_O_2_, the ratio of JC-1 aggregates to monomer in LHON-SkQ1-H_2_O_2_ showed a considerable increase compared with that in LHON-DMSO-H_2_O_2_, indicating that SkQ1 promoted mitochondrial membrane stability (Figure 4A lower part and Figure 4B right part).

#### 3.4.2. SkQ1 Reduced the Degree of MPTP Opening

Decreased MMP leads to elevated mitochondrial permeability. To further validate the effect of SkQ1 on mitochondrial function, we used the Mitochondrial Permeability Transition Pore (MPTP) assay kit to estimate the degree of permeability. The results exhibited that the degree of permeability was decreased in LHON-SkQ1 compared to LHON-DMSO (Figure 4C upper part and Figure 4D left part). Then, we added 200 μM H_2_O_2_ for 45 min to stimulate IFBs. The degree of permeability in LHON-IFBs was ameliorated with SkQ1 pretreatment compared with that in LHON-DMSO-H_2_O_2_ (Figure 4C lower part and Figure 4D right part).

#### 3.4.3. SkQ1 Improved Mitochondrial Respiratory Function

We further explored whether SkQ1 improves the mitochondrial respiratory function of LHON-IFBs. The results of Seahorse cell energy metabolism analysis showed that the OCR of LHON-IFBs pretreated with SkQ1 was increased compared with that of untreated LHON-IFBs, and in detail, basal oxygen consumption and maximal oxygen consumption were elevated, which may be associated with enhancing oxidative phosphorylation in mitochondria (Figure 4E,F). These results indicated that SkQ1 effectively improved the mitochondrial respiratory function of LHON-IFBs.

### 3.5. Network Pharmacology and Molecular Docking Validation

#### 3.5.1. Screening Potential Protein Targets of SkQ1

To explore the protein mechanism of SkQ1 in LHON, network pharmacology was employed to identify potential targets of SkQ1. A total of 111 candidate targets of SkQ1 were screened from various databases including PubChem, STITCH, SEA (similarity ensemble approach), TargetNet, and Swiss Target Prediction databases. Additionally, a total of 200 target genes related to LHON were collected from TTD (Therapeutic Target Database), OMIM, GeneCards, and DisGeNET databases. Using the Venn diagram, the overlapping proteins, including CAT, SOD2, PPARGC1A, PINK1, VDAC1, MFN1, TP53, MFN2, GSR, IL1B, and NDUFS4, were defined as the potential targets of SkQ1 (Figure 5A).

#### 3.5.2. Constructing a PPI Network and Topological Analyses

The PPI network was constructed using the String database (https://cn.string-db.org/, accessed on 6 April 2022) to gain access to the underlying relationships and biological processes of potential targets. Subsequently, the network was visualized using Cytoscape 3.9.1 software. The PPI network contained 11 nodes and 38 edges (Figure 5B). To obtain the hub targets of SkQ1 among potential targets, a topological analysis of the PPI network was conducted. The diagrams revealed that the top five targets, including CAT, SOD2, PPARGC1A, PINK1, and VDAC1, were identified as hub targets (degree > 8, betweenness > 9, closeness > 5.92, Figure 5C).

#### 3.5.3. GO and Enrichment Analysis and Molecular Docking Simulation

Gene ontology (GO, https://www.geneontology.org/, accessed on 13 May 2022) enrichment and signaling pathway enrichment analysis of target proteins were performed by Metascape (https://www.metascape.org, accessed on 25 April 2022) and DAVID (https://david.ncifcrf.gov/, accessed on 26 April 2022). GO enrichment analysis identified a total of 125 biological processes (BC), 16 cellular compositions (CC), and five molecular functions (MF). The top five entries were selected and ranked by values from BP, CC, and MF, respectively. As shown in Figure 5D, the top GO items were implicated in intracellular oxidative stress, mitochondria, and redox activity. In addition, REACTOME pathway enrichment analysis revealed 19 significantly enriched signaling pathways (*p* < 0.01, FDR < 0.05), primarily implicated in mitochondrial autophagy and oxidative stress (Figure 5E). Moreover, molecular docking analysis and 3D structure examination indicated that PINK1, VDAC1, SOD2, and CAT exhibited stronger binding affinity towards SkQ1 (Figure 5F). Proteins SOD2 and CAT play roles in the oxidative stress pathway, while proteins PINK1 and VDAC1 are constituents of the mitophagy process. Therefore, we presumed that these four proteins were potential targets of SkQ1, and oxidative stress may lead to mitophagy.

### 3.6. SkQ1 Interacted with PINK1

To investigate the interaction of SkQ1 with key target proteins PINK1 and SOD2 in vitro, we used cellular thermal shift assay (CETSA) by treating LHON-IFBs with SkQ1 for 24 h and lysing them after heating them for a certain period of time at different temperatures; proteins bound to the drug would be less prone to denaturation and precipitation due to the increased thermal stability, and finally the target proteins within the supernatant were detected by Western blot. The results showed that in the samples treated with SkQ1, the thermal stability of PINK1 and SOD2 gradually increased with increasing temperature compared with the control group (Figure 6A,B). Thus, we hypothesized that SkQ1 could interact with PINK1 and SOD2 to regulate mitophagy and oxidative stress.

### 3.7. SkQ1 Inhibited PINK1/PRKN-Mediated Mitophagy

To further verify the effect of SkQ1 on key target proteins, we detected the protein expression using Western blot. The results showed that the expression of PINK1, PRKN (same as Parkin), LC3B, and SOD2 were down-regulated in LHON-IFBs after SkQ1 (20 nM) treatment compared with the control group, regardless of H_2_O_2_ (200 μM/6 h)-induced or non-induced (Figure 7A–D). The difference in SOD2 between LHON-IFBs and control group was not significant (Figure 7E). The protein level of VDAC1 remained unchanged (Figure 7F). Verification was also performed in vivo in HEK293T and RPE-1 cells, and the results showed that the expression of PINK1 and LC3B were down-regulated in both cell lines after SkQ1 (20 nM) treatment compared with the control (Figure 7G,H). The above results indicated that SkQ1 could inhibit PINK1/PRKN-mediated mitophagy by down-regulating the expression of PINK1, PRKN, and LC3B.

Taken together, these findings indicate that when mitochondrial respiratory chain complex I dysfunction occurs, resulting in electron leakage and increased ROS, SkQ1 could scavenge excessive ROS and interact with and down-regulate the expression of PINK1 to inhibit PINK1/PRKN-mediated mitophagy, thereby protecting mitochondrial function and maintaining mitochondrial homeostasis (Figure 7I).

## 4. Discussion

LHON is a mitochondrial DNA point mutation disease [20]. Most of the mutations affect a single subunit of mitochondrial NADH dehydrogenase, leading to dysfunction of complex I in the electron transport chain [21]. Mitochondria damage contributes to ROS increasing in RGCs that critically depend on adequate ATP supply, which may be a hazardous factor in visual system degeneration [22,23]. Excess ROS in turn further damage mitochondrial DNA, which has a high mutation rate due to lack of introns, which is a vicious circle [24]. Therefore, antioxidants become one of the crucial intervention strategies. We found that SkQ1, targeting mitochondrial ROS, could significantly protect mitochondrial function and be a potential medicine to ameliorate optic neuropathy.

While the most common clinical feature of LHON is optic nerve atrophy, the affected tissue is not easily available for research. Thus, a variety of alternative cells and tissues have been tested [25]. We collected skin fibroblasts from two LHON donors, LHON-I and LHON-II, both carrying the mutation m.11778G > A, and healthy volunteers distributed over a similar age range (25–35 years) to investigate the therapeutic molecular mechanisms of SkQ1 [18,19].

Firstly, we explored the appropriate concentration of SkQ1 in skin fibroblasts derived from LHON patients. Our experimental findings showed that SkQ1 had no adverse effects on cell proliferation at nanomolar concentration, while a toxic effect of SkQ1 was observed at micromolar concentrations, similarly to Antonenko et al.’ results [13]. Moreover, we observed no detrimental effect on cell viability after long-term SkQ1 treatment, which was consistent with the results of other researchers [11,12].

Elevated ROS is considered a primary pathological process of complex I dysfunction in LHON [26], and disruption of redox homeostasis contributes to mitochondrial dysfunction [27], which in turn leads to retinal neuronal degeneration and ultimately RGC loss [5,7]. Based on ROS detection experiments, the images indicated that SkQ1 attenuated ROS both in cytoplasm and mitochondria, which was consistent with the results of other scholars [28,29]. Some researchers have suggested the antioxidant mechanisms of SkQ1: (I) Ion pairing of the cation SkQ1+ with the free fatty acid anion leads to uncoupling; (II) ability of SkQ1H_2_ to interact with lipo peroxyl radicals; (III) interference with electron flow at the ubiquinone binding site within complex III, involving the reduction of SkQ1 to SkQ1H2 by ubiquinone [30,31]. In addition, we found that SkQ1 ameliorated MMP depolarization and reduced the degree of MPTP opening, which indicated that SkQ1 could improve mitochondrial function.

Further, based on the property that small-molecule compounds bind easily to protein, we posited that ligand SkQ1 could combine with mitochondria-associated protein receptors [32]. Thus, we opted to utilize the network pharmacology and molecular docking to systematically explore the underlying protein mechanism of SkQ1 [33]. Thus, we ended up with five hub targets: PPARGC1A, CAT, SOD2, PINK1, and VDAC1, and these targets were major participants in two signaling pathways: intracellular oxidative stress, and PINK1/PRKN-mediated mitochondrial autophagy. And data implied that SkQ1 bound steadily to these hub target proteins.

We conducted Western blot to validate the results of the network analysis in vitro. Protein CAT and SOD2 play key roles in the antioxidant pathway [34]. It was noted that the protein level of SOD2 returned to normal values with SkQ1 pretreatment, although the difference was not significant. Previous studies have shown that gene transcript levels and enzyme activities of SOD and CAT in the heart and blood of hyperglycemic rats were reduced after administration of SkQ1 [35]. However, according to the data provided by the literature, the protein expression of CAT is relatively low in skin and eye tissues, and it is situated in the cytoplasm rather than in the mitochondria, and the binding of SkQ1 to CAT was not validated by Western blot [36,37].

Mitophagy is required for mitochondrial quality control (MQC) maintenance, an evolutionarily conserved mechanism [38]. PINK/PRKN-mediated mitophagy is one of the mitophagy pathways. In this process, PINK1 as a sensor of mitochondrial depolarization and recruits PRKN to clear damaged mitochondria. Additionally, this pathway is also involved in regulating mitochondrial dynamics through preventing defective mitochondria from fusing [39]. In our study, when ROS increased, causing mitochondrial membrane potential depolarization, and mitophagy was triggered subsequently, SkQ1 interacted with PINK1 and inhibited the PINK1/PRKN-mediated pathway, effectively preventing excessive mitophagy and maintaining mitochondrial fitness.

It is well-known that both ROS and mitophagy play the role of a “double-edged sword” in cells [40]. On the one hand, ROS overload causes oxidative damage to proteins and lipid molecules, which underlies the onset of aging and many metabolic and neurodegenerative diseases [41]. On the other hand, moderately increased ROS can serve as a signal to induce autophagy and other cell survival mechanisms [39]. Secondly, moderate mitophagy has a positive effect on the reduction of ROS produced by damaged mitochondria and constitutes a negative feedback regulatory mechanism to reduce intracellular oxidative damage [42]. Excessive mitophagy may degrade normal mitochondria and cause further cellular damage [43,44].

In conclusion, our data further supported that SkQ1 could scavenge mitochondrial ROS to protect mitochondrial function. Meanwhile, we hypothesized that SkQ1 could interact with PINK1 and regulate PINK1/PRKN-mediated mitophagy to maintain mitochondrial homeostasis in skin fibroblasts carrying the m.G11778A mutation. We believed that SkQ1 would be an efficient and low-toxicity treatment for LHON.

## 5. Conclusions

We employed a combination of network pharmacology and in vitro experimental validation to investigate the effects and molecular mechanism of SkQ1 in the preservation of mitochondrial function using skin fibroblasts derived from LHON patients as well as HEK293T. There are three main highlights of the findings. First, we provided that SkQ1 could scavenge mitochondrial ROS and stabilize mitochondrial membrane potential. Second, we found, through network pharmacology and molecular docking, that the key targets of SkQ1 are SOD2 and PINK1. Third, we validated that SkQ1 interacted with PINK1 and inhibited PINK1/PRKN-mediated mitophagy to maintain mitochondrial homeostasis. Our findings suggest that by regulating PINK1/PRKN-mediated mitophagy, SkQ1 preserves mitochondrial function in LHON cells. The data indicate that SkQ1 may be a novel therapeutic intervention to prevent the progression of LHON.

## Figures and Tables

**Figure 1 biomedicines-12-02020-f001:**
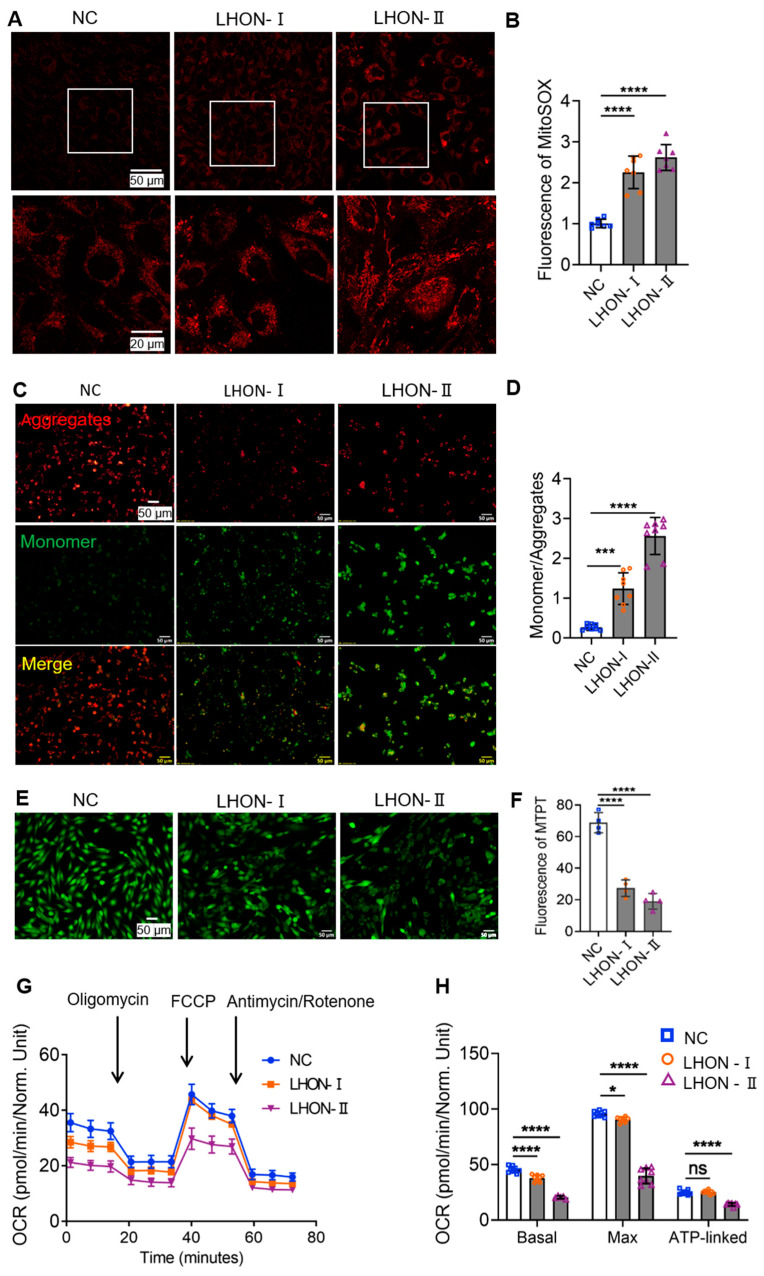
LHON-IFBs presented excessive superoxide and mitochondrial dysfunction. (**A**,**B**) Increased superoxide in the mitochondria of LHON-IFBs. (**C**,**D**) The ratio of monomer to aggregates is inversely proportional to the MMP. (**E**,**F**) Increased mitochondrial membrane permeability of LHON-IFBs. (**G**,**H**) Decreased OCR in the mitochondria of LHON-IFBs. * *p* ≤ 0.05, *** *p* ≤ 0.001, **** *p* ≤ 0.0001, ns = no significance, *n* ≥ 4.

**Figure 2 biomedicines-12-02020-f002:**
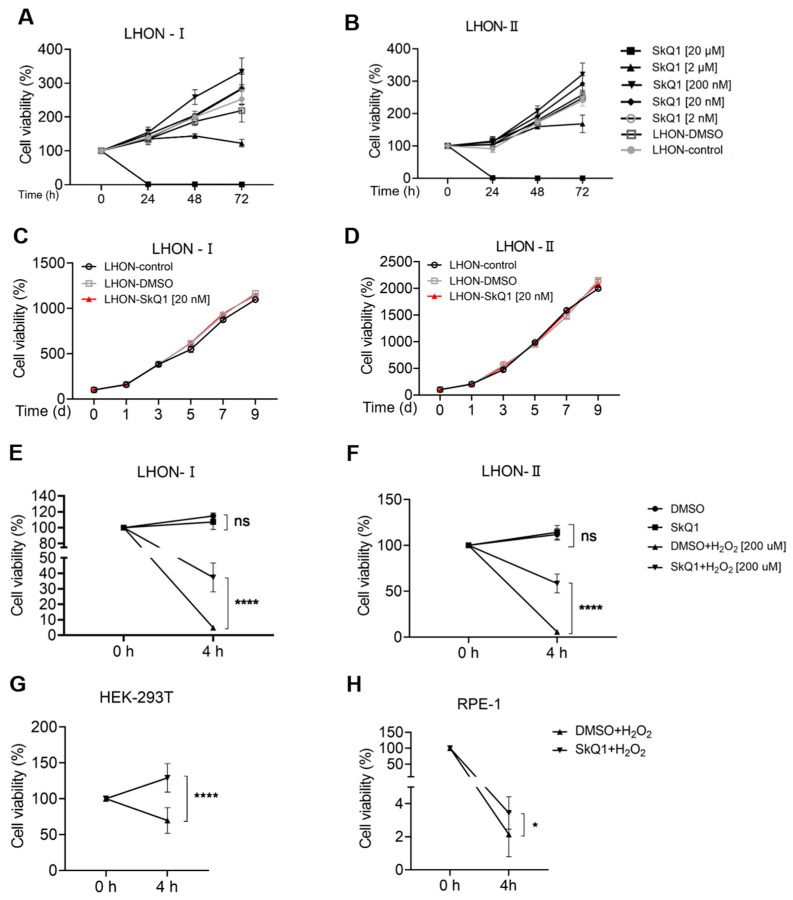
Drug safety was evaluated by CCK8 assay. (**A**,**B**) Cell viability was unaffected by the SkQ1 concentration from 2 nM to 200 nM. (**C**,**D**) LHON-IFBs were cultured with 20 nM SkQ1 for 9 days. The cell viability was detected on 1st, 3rd, 5th, 7th, and 9th day, respectively, using CCK8. (**E**,**F**) By adding H_2_O_2_ (200 μM) for 4 h, the viability of cells pretreated with SkQ1 was significantly higher than that of the control group. (**G**,**H**) By adding H_2_O_2_ (200 μM) for 4 h, the viability of HEK293T and RPE-1 pretreated with SkQ1 was also significantly higher than that of the control group. * *p* ≤ 0.05, **** *p* ≤ 0.0001, ns = no significance, *n* ≥ 4.

**Figure 3 biomedicines-12-02020-f003:**
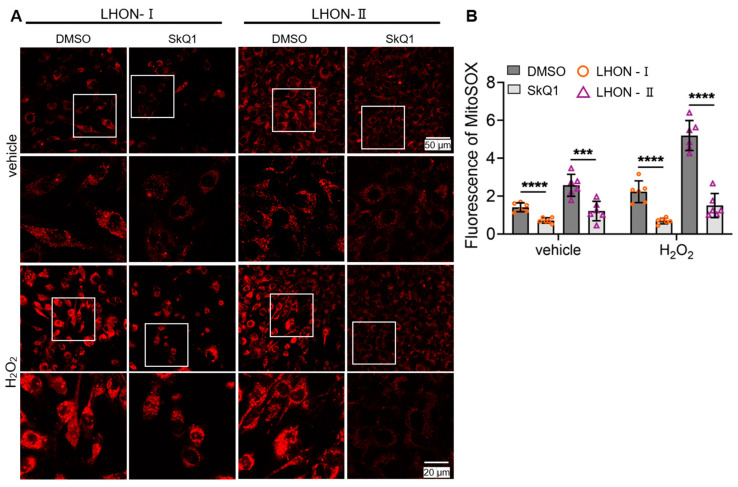
Effect of SkQ1 in scavenging superoxide. (**A**,**B**). The level of mitochondrial superoxide decreased after 7-day treatment with 20 nM SkQ1. *** *p* ≤ 0.001, **** *p* ≤ 0.0001, *n* ≥ 4.

**Figure 4 biomedicines-12-02020-f004:**
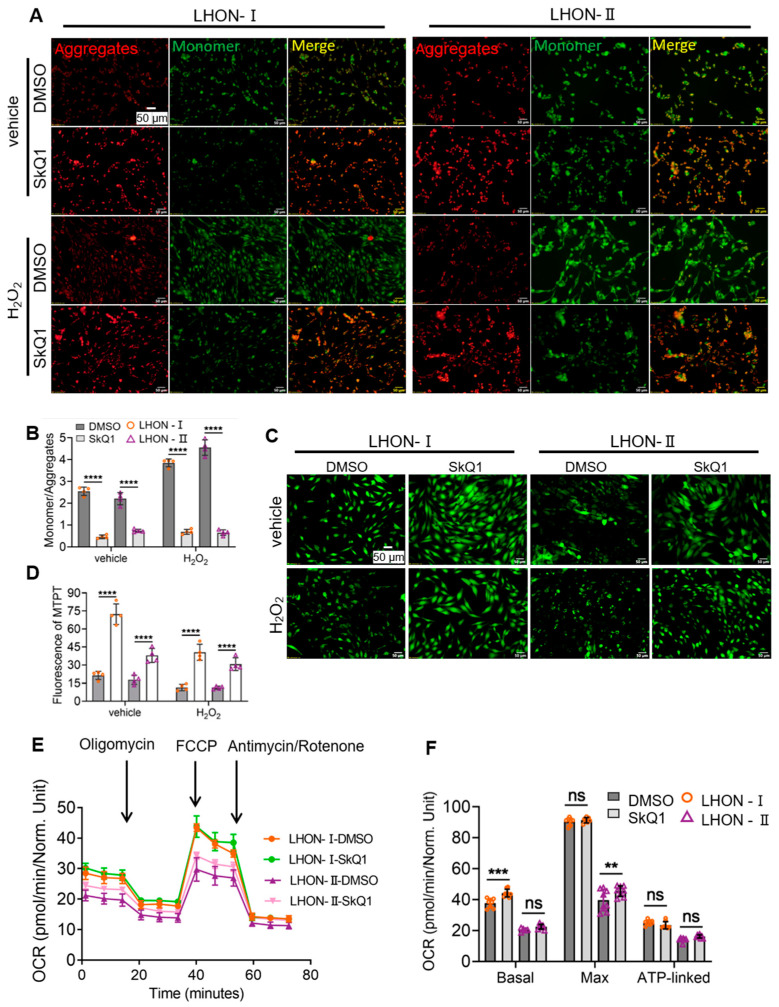
Effect of SkQ1 in protecting mitochondrial function. (**A**,**B**) SkQ1 improved the MMP of LHON-IFBs. Green fluorescence and red fluorescence represent JC-1 monomer and JC-1 aggregate, and a larger ratio of JC-1 monomer to aggregates indicated a greater proportion of depolarization of the MMP. (**C**,**D**) Mitochondrial membrane permeability is inversely proportional to the intensity of green fluorescence in mitochondria. (**E**,**F**) Mitochondrial OCR was increased in LHON-IFBs pretreated with SkQ1. ** *p* ≤ 0.01, *** *p* ≤ 0.001, **** *p* ≤ 0.0001, ns = no significance, *n* ≥ 4.

**Figure 5 biomedicines-12-02020-f005:**
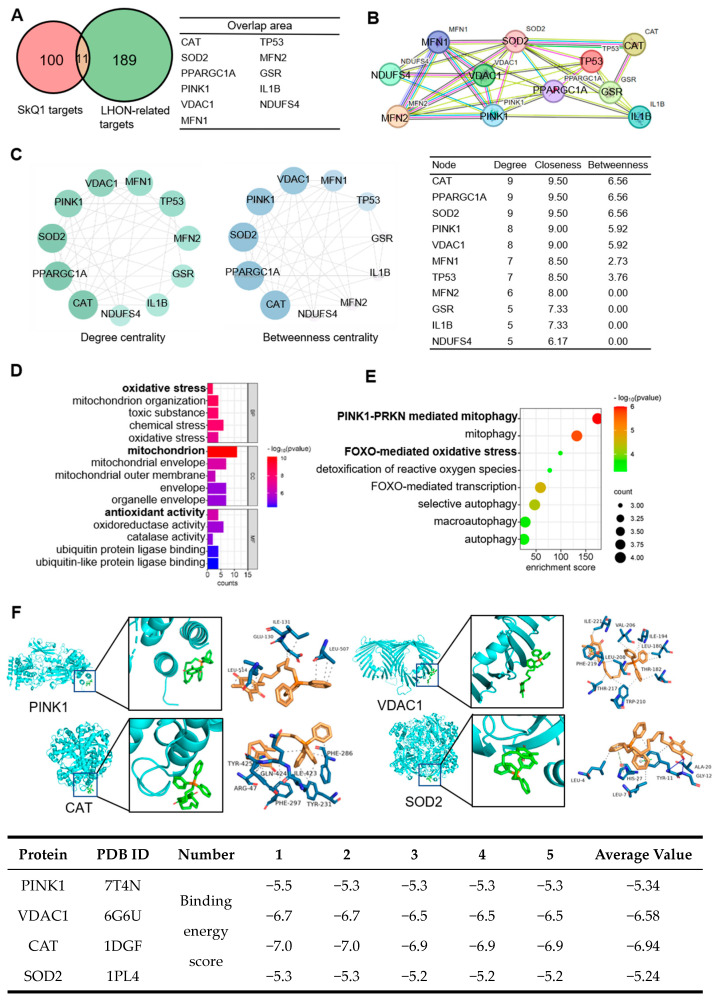
Potential target analysis and molecular docking. (**A**) Venn diagram of common SkQ1 (drug) and LHON (disease) targets. (**B**) PPI networks of potential targets analyzed by STRING 11.5. (**C**) Topological analysis of potential targets. (**D**) Gene ontology enrichment analysis. The bars list the top five biological processes (BP), cellular components (CC), and molecular functions (MF) according to the parameter -lg*P*. (**E**) Enrichment analysis of signaling pathways. The top eight REACTOME pathways were listed in the bubble chart according to the parameters *p* < 0.01 and FDR < 0.05. (**F**) Molecular docking diagrams of potential targets and the binding energy score < −5 were considered to indicate tighter binding of the receptor to the SkQ1.

**Figure 6 biomedicines-12-02020-f006:**
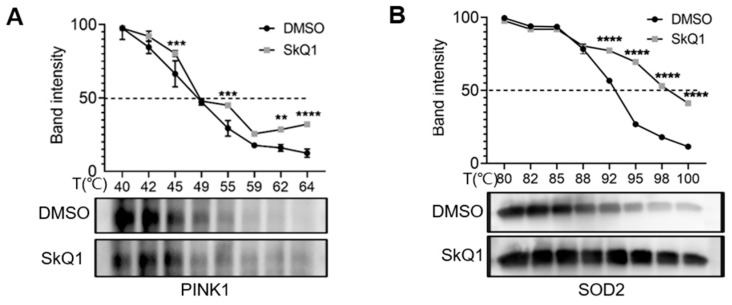
The interaction of SkQ1 with PINK1 and SOD2 was evaluated by CETSA assay. (**A**,**B**) The results showed that in the samples treated with SkQ1, the thermal stability of PINK1 and SOD2 increased with increasing temperature as compared to control. ** *p* ≤ 0.01, *** *p* ≤ 0.001, **** *p* ≤ 0.0001, *n* ≥ 3.

**Figure 7 biomedicines-12-02020-f007:**
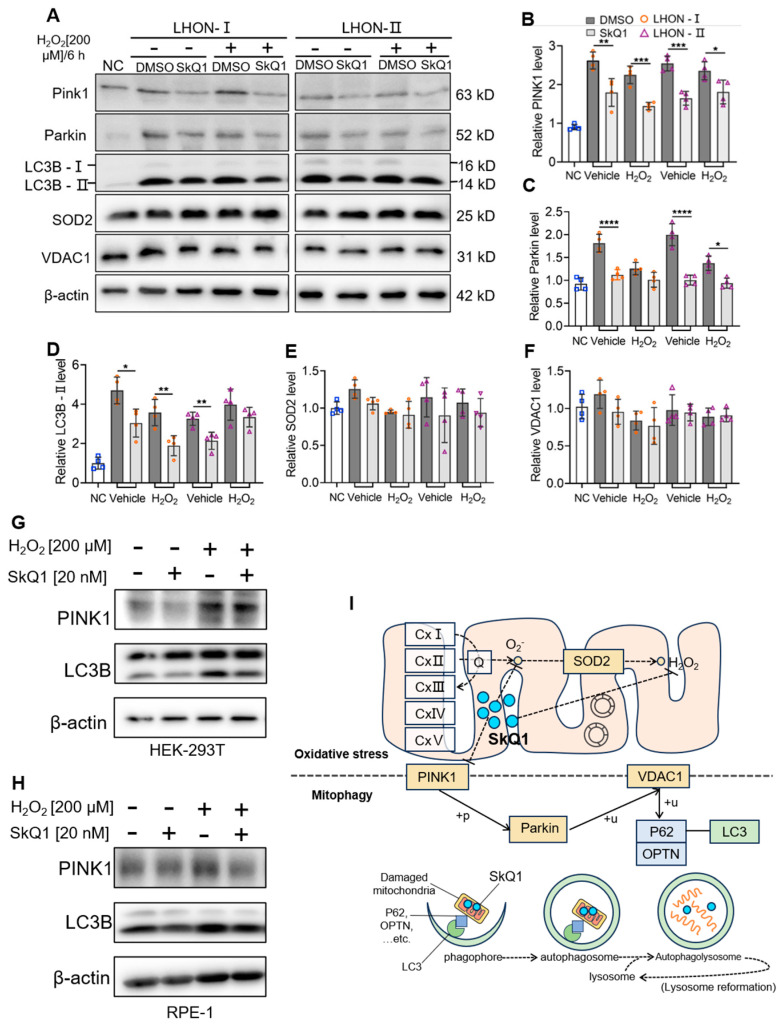
SkQ1 regulated PINK1/PRKN-mediated mitophagy. (**A**–**F**) LHON-IFBs were pretreated with SkQ1 (20 nM) and DMSO for 7 days, followed by treatment with 200 μM H_2_O_2_ for 6 h. The expressions of PINK1, PRKN, LC3B, SOD2, VDAC1, and β-actin protein were detected by Western blot. β-actin was used as a reference, *n* ≥ 4. (**G**,**H**) The expression levels of PINK1 and LC3B in HEK293T and RPE-1 cells were consistent with those of LHON-IFBs, * *p* ≤ 0.05, ** *p* ≤ 0.01, *** *p* ≤ 0.001, **** *p* ≤ 0.0001, *n* ≥ 3. (**I**) SkQ1 could be involved in regulating oxidative stress and PINK1/PRKN-mediated mitophagy pathway.

## Data Availability

The original contributions presented in the study are included in the article, further inquiries can be directed to the corresponding author.

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
