# Peer review of "Preservation of Mitochondrial Function by SkQ1 in Skin Fibroblasts Derived from Patients with Leber’s Hereditary Optic Neuropathy Is Associated with the PINK1/PRKN-Mediated Mitophagy"

_biomedicines, 2024, doi:10.3390/biomedicines12092020_

Round 1

Reviewer 1 Report (Previous Reviewer 2)

Comments and Suggestions for Authors

No more comment to add

Comments on the Quality of English Language

Just checking some sentences. 

Author Response

Comments 1: Just checking some sentences.

Response 1: Thank you for pointing this out. We have corrected some irregular expressions in this manuscript. (such as, page 2, line 18, Delete “we posit” .page 2, line 14 “in improving” changed to “to improve”, page 4 , line 22, “CO2” changed “CO2”, et al)

Reviewer 2 Report (Previous Reviewer 3)

Comments and Suggestions for Authors

the re-submission by Xu et al addresses some of the issues but still the data and arguments presented are not strong enough to support their conclusion - not only their experimental models are limited but also the assays conducted are inappropriate. Specifically mitoSOX (and DHE, not DCF) data must be analysed on both the blue and red channels to obtain a ratiometric ratio in order to rule out any difference in the dye uptake.

Agilent has published their technical guidance in analysing fibroblasts (TB_Fibroblasts.pdf (agilent.com)  that differs from the conditions the authors have used.  more importantly it is known that because of the high level of HEPES in the SeaHorse medium, it is impossible to conduct BCA post-run.

Author Response

Comments 1: The data and arguments presented are not strong enough to support their conclusion - not only their experimental models are limited but also the assays conducted are inappropriate.

Response 1: Thank you very much for pointing out the shortcomings of the manuscript and giving valuable suggestions. We agree that skin fibroblasts from LHON patients does not fully represent the optic ganglion cells. At this point, we have reviewed the literature and learned that the new models at present are the iPSCs (Induced pluripotent stem cells) and animals. Riazifar H, et al reported that a single chemical (DAPT) can induce PAX6/RX-positive stem cells to undergo differentiation into functional RGCs [1]. Besides, The other is a rat model injected with rotenone microspheres, which exhibits the most common features of LHON, mainly the optic nerve and retina atrophy. Bahr T et al reported that fibroblasts are thought to have more similarities with RGCs and can be cultured from skin biopsy. Like blood cells, an advantage of using fibroblasts is the representation of the nuclear genome of an LHON-afflicted patient. Cultured fibroblasts have been used to study protein expression, ATP production, and the effects of environmental factors in LHON, as well as respiration and ETC complex function [2]. Danese A et al also used fibroblasts derived from LHON patients and found that in basal conditions, compared with control cells, fibroblasts from LHON patients, considered the most severe for biochemical dysfunction [3]. In this study, we also experimentally confirmed the presence of mitochondrial dysfunction in fibroblasts from G11778A LHON patients compared to healthy controls

[1] Riazifar H, Jia Y, Chen J, Lynch G, Huang T. Chemically induced specification of retinal ganglion cells from human embryonic and induced pluripotent stem cells. Stem Cells Transl Med. 2014;3(4):424-432.

[2] Bahr T, Welburn K, Donnelly J, Bai Y. Emerging model systems and treatment approaches for Leber's hereditary optic neuropathy: Challenges and opportunities. Biochim Biophys Acta Mol Basis Dis. 2020 Jun 1;1866(6):165743).

[3] Danese A, Patergnani S, Maresca A, et al. Pathological mitophagy disrupts mitochondrial homeostasis in Leber's hereditary optic neuropathy. Cell Rep. 2022;40(3):111124.

Comments 2: mitoSOX (and DHE, not DCF) data must be analysed on both the blue and red channels to obtain a ratio metric ratio in order to rule out any difference in the dye uptake.

Response 2: Thank you for pointing this out. We agree with this comment. The manuscript has been revised accordingly to clarify the above concerns. We used the fluorescence values of the negative control (NC) group as an internal reference and homogenized all data to exclude differences due to dye uptake.We have supplemented the Methods section with information on the fluorescence statistics of the MitoSOX experiment and have recalculated the results of this experiment to exclude differences due to dye uptake, which have been updated (page 3, line 33, page6, figure1D, page 8, figure3D).

Comments 3. Agilent has published their technical guidance in analysing fibroblasts (TB_Fibroblasts.pdf (agilent.com) that differs from the conditions the authors have used. it is known that because of the high level of HEPES in the SeaHorse medium, it is impossible to conduct BCA post-run.

Response 3: Thank you for your suggestion. We followed the instructions on the Agilent website and did not use HEPES buffer in this experiment. In addition, we also found literature on fibroblasts on Agilent's website, which reported the use of BCA to detect protein concentration. Habarou F, et al have proposed that Protein concentrations in each well were determined with the BCA method (Pierce) in cell lysates after the measurement [4].

[4] Habarou F, Hamel Y, Haack TB, et al. Biallelic Mutations in LIPT2 Cause a Mitochondrial Lipoylation Defect Associated with Severe Neonatal Encephalopathy. Am J Hum Genet. 2017;101(2):283-290.

Round 2

Reviewer 2 Report (Previous Reviewer 3)

Comments and Suggestions for Authors

the authors are yet to address my comments.  It is more the assay choice and data interpretation that falls short in supporting their conclusion.  personally I'd rather taking out the DCF data - it doesn't add support but confuses the field to carry on using this assay.  the ratiometric calculation from mitoSOX allows per cell, in the same cell, correlation of oxidised product by the reduced dye uptake.  finally BCA is known incompatible to SeaHorse medium since it contains a high level of HEPES that causes extremely high backgr, therefore people either image the cells or assay DNA content for cell numbers.  I cannot be convinced their data is not artifact.

I'm happy to re-consider this submission if the authors are to remove their DCF data, re-analyse/re-run their DHE data and reliably assay the cell numbers in their SeaHorse runs.  these represent the standard requirements in the field in studies of this kind.

Author Response

Comments 1: personally I'd rather taking out the DCF data - it doesn't add support but confuses the field to carry on using this assay.

Response 1: Thank you very much for pointing out the shortcomings of the manuscript and giving valuable suggestions. We agree that the DCF data is not strong enough to support the conclusion, so we have taken out the DCF data. 

Comments 2: the ratiometric calculation from mitoSOX allows per cell, in the same cell, correlation of oxidised product by the reduced dye uptake.

Response 2: Thank you for pointing this out. We agree with this comment. The manuscript has been revised accordingly to clarify the above concerns. We used the lasso tool of ImageJ software to outline each group of six fluorescent cells to be tested and three background objects, which were used for subsequent deduction of background values. Then, we calculated and analyzed mean intensity of MitoSOX fluorescence. Calibrated Cumulative Optical Density = Cumulative Optical Density - Cell Area to be Measured × Average Background Optical Density. We have supplemented the Methods section with information on the fluorescence statistics of the MitoSOX experiment and have recalculated the results of this experiment to exclude differences due to dye uptake, which have been updated (page 3, line 29, page 6, figure1A-B, page 8, figure3A-B).

Comments 3. BCA is known incompatible to SeaHorse medium since it contains a high level of HEPES that causes extremely high background, therefore people either image the cells or assay DNA content for cell numbers.

Response 3: Thank you for pointing this out. We agree with this comment. The manuscript has been revised accordingly to clarify the above concerns. We learned that the XF medium at pH 7.4 contained a small amount of HEPES buffer, and we normalized the data using cell imaging combined with cell counting. We modified the methodology regarding the normalization of the seahorse experiment and recalculated the results of this experiment, which have been updated (page 4, line 19, page 6, figure1 G-H, page 9, figure4 E-F).

Round 3

Reviewer 2 Report (Previous Reviewer 3)

Comments and Suggestions for Authors

clearly the authors have made an effort to address the points, which by large are acceptable.  re. mitoSOX the proper way to run ratiometric analysis is to image the blue and red fluorescence and express them in ratio but that may involve the authors to re-do the expt.

This manuscript is a resubmission of an earlier submission. The following is a list of the peer review reports and author responses from that submission.

Round 1

Reviewer 1 Report

Comments and Suggestions for Authors

This present study investigated the effects and molecular mechanisms of SkQ1, a rechargeable mitochondria-targeted antioxidant, in the preservation of mitochondrial function in skin fibroblasts derived from patients with Leber's hereditary optic neuropathy (LHON). The study found that SkQ1 could reduce reactive oxygen species (ROS) production and stabilize the mitochondrial membrane. Through network pharmacology and molecular docking analyses, the authors identified superoxide dismutase 2 (SOD2) and PTEN-induced kinase 1 (PINK1) as key targets of SkQ1, which play crucial roles in redox regulation and mitophagy, respectively. Importantly, SkQ1 interacted with PINK1 and downregulated its expression, thereby regulating PINK1/PRKN-mediated mitophagy and preserving mitochondrial function in LHON fibroblasts. The authors suggest that SkQ1 may be a novel therapeutic intervention to prevent the progression of LHON by modulating mitochondrial homeostasis. However, there are several insufficiencies in the publication of this study in Biomedicines, both in terms of the experimental method and interpretation of the results.

Commnets

1. What exactly does the decrease in MPTP fluorescence signal in Figure 1E mean? In general, increased MPTP activity induces depolarization of the mitochondrial membrane potential, leading to mitochondrial dysfunction. Please present the mitochondrial JC-1 analysis membrane potential results directly related to this data by comparing them in the NC, LHON-1, and LHON-2 groups. In addition, Figure 4A and B also do not include a normal control group, so it is not possible to know whether the membrane potential of LHON-I and II was actually depolarized compared to normal. Please also add the results of treatment of SkQ1 in NC.

2. In the 2.2 Drug safety of SkQ1 section, the following description does not match the figure. Please fix it. “The graph displayed that there was a gradual rise in the LHON-IFBs with treatment concentrations of SkQ1 ranging from 2 nM to 200 nM, despite the cell viability being inhibited beyond 2 μM SkQ1 (Figure 1G). Besides, figure 1H showed that there was continual growth in cell viability during the period of treatment with 20 nM SkQ1 for 9 days. It revealed no significant difference in cell viability among the three groups, indicating that 20 nM was the available concentration with low cytotoxicity.”

3. The results for Figures 2E and F are not presented in the text. Please explain the results, including the reason why authors used the corresponding cell line rather than LHON-1 and 2 cells in the cell viability experiment by H2O2 treatment.

4. In Figure 3A and D, there are no comparison results of H2O2 and SkQ1 effects in Normal Control. Please add

5. In Figure 6B, the SOD2 Western quantitative graphs of DMSO and SkQ1 appear almost overlapping, and it is difficult to tell whether there is actually a statistical difference.

6. In Figure 7, it appears that H2O2 was used as a positive control to induce mitophagy, but in LHON-I and II, mitophagy was not induced at all by H2O2.

7. In Figure 4E, SkQ1 increases mitochondrial oxygen consumption in LHON-II cells, but an accurate explanation is needed.

Comments on the Quality of English Language

none

Author Response

Comments 1: What exactly does the decrease in MPTP fluorescence signal in Figure 1E mean? In general, increased MPTP activity induces depolarization of the mitochondrial membrane potential, leading to mitochondrial dysfunction. Please present the mitochondrial JC-1 analysis membrane potential results directly related to this data by comparing them in the NC, LHON-1, and LHON-2 groups. In addition, Figure 4A and B also do not include a normal control group, so it is not possible to know whether the membrane potential of LHON-I and II was actually depolarized compared to normal. Please also add the results of treatment of SkQ1 in NC.

Response 1: Thank you for pointing this out. We agree with this comment. Therefore, l have added an explanation of this concept in the manuscripts (Page 5, results 2.1, paragraph 2, line 8)and supplemented the experimental proofs with the JC-1 experimental results in the text (Page 6, Figure 1E-F).

Comments 2: In the 2.2 Drug safety of SkQ1 section, the following description does not match the figure. Please fix it. “The graph displayed that there was a gradual rise in the LHON-IFBs with treatment concentrations of SkQ1 ranging from 2 nM to 200 nM, despite the cell viability being inhibited beyond 2 μM SkQ1 (Figure 1G). Besides, figure 1H showed that there was continual growth in cell viability during the period of treatment with 20 nM SkQ1 for 9 days. It revealed no significant difference in cell viability among the three groups, indicating that 20 nM was the available concentration with low cytotoxicity.”

Response 2: Agree. We have re-matched the description to the images (Page 7).

Comments 3. The results for Figures 2E and F are not presented in the text. Please explain the results, including the reason why authors used the corresponding cell line rather than LHON-1 and 2 cells in the cell viability experiment by H2O2 treatment.

Response 3: Agree. We have re-matched the description to the images (Page 7). We performed the same experiments on LHON-1 and LHON-2 cells, and latest experimental results have been presented in the manuscript (Page7, Figure 2E-F).

Comments 4. In Figure 3A and D, there are no comparison results of H2O2 and SkQ1 effects in Normal Control. Please add

Response 3: Thank you for pointing this out. Because previous experimental results showed no difference between the DMSO group and the NC group, and CCK8 results also presented no statistically significant difference in cell viability between long-term treatment in the DMSO and NC groups, here we only put the results of the DMSO group.

Comments 5. In Figure 6B, the SOD2 Western quantitative graphs of DMSO and SkQ1 appear almost overlapping, and it is difficult to tell whether there is actually a

statistical difference.

Response 3: Agree. We have supplemented the experimental proofs with the latest experimental results in the text (Page 12, results 2.6, Figure 6B).

Comments 6. In Figure 7, it appears that H2O2 was used as a positive control to induce mitophagy, but in LHON-I and II, mitophagy was not induced at all by H2O2.

Response 3: Thank you for pointing this out. According page 13, Figure 7A displayed that the expression of PINK1, Parkin and LC3B was significantly increased by the addition of H2O2.

Comments 7. In Figure 4E, SkQ1 increases mitochondrial oxygen consumption in LHON-II cells, but an accurate explanation is needed.

Response 3: Agree. l have added an explanation of this result in the manuscripts (Page 9, results 2.4.3, line 5)

Reviewer 2 Report

Comments and Suggestions for Authors

I like this manuscript suggesting SkQ1 as a novel therapeutic approach for LHON progression but I have some concerns related to mitophagy:

1) Figure 1. Authors need to validate genetic modification on fibroblast, it is possible to sequence fibroblast from patients?

2) Figure 4 panel F, ATP production is not the correct way to report, I suggest changing it in the ATP linked

3) Paragraph 2.7 line 372: Since mitophagy can be regulated also by mitofusin2, (Li J. Front Cell Dev Bio)  It is possible to validate mitofusin ubiquitination and parkin translocation on Mfn2. 

4)Since mitophagy is linked to the mitochondria fission/fusion process, the author needs to show mitochondria confocal pictures in LHON patients. 

5)Discussion: Line 456 authors need to add more sentences related to mitochondria dynamism, there is new evidence on mitochondria fusion/ fission/ mitophagy and mitochondria clumping (Franco A. Life 2022)

Comments on the Quality of English Language

None 

Author Response

Comments 1: Figure 1. Authors need to validate genetic modification on fibroblast, it is possible to sequence fibroblast from patients?

Response 1: We are grateful to the reviewers for the suggestions, and we have supplemented this section in the discussion (Page 14, line 10)[1,2]. The main purpose of this study was to further the investigation of novel treatment strategies for LHON based on the previous work, so the preliminary genetic examination was not a key element of our focus, and we do not think that absence of this part will have much impact on the focus of this study.

[1] Yao S, Zhou Q, Yang M, Li Y, Jin X, Guo Q, Yang L, Qin F, Lei B. Multi-mtDNA Variants May Be a Factor Contributing to Mitochondrial Function Variety in the Skin-Derived Fibroblasts of Leber's Hereditary Optic Neuropathy Patients. Front Mol Neurosci. 2022 Jul 13;15:920221.

[2] Zhou Q, Yao S, Yang M, Guo Q, Li Y, Li L, Lei B. Superoxide dismutase 2 ameliorates mitochondrial dysfunction in skin fibroblasts of Leber's hereditary optic neuropathy patients. Front Neurosci. 2022 Aug 9;16:917348.

Comments 2: Figure 4 panel F, ATP production is not the correct way to report, I suggest changing it in the ATP linked

Response 2: Thank you for pointing this out. We agree with this comment. We have made corrections in the manuscript (Page 6, Figure 1J, Page 10, Figure 4F, Page 9, results 2.4.3, line 4).

Comments 3 Paragraph 2.7 line 372: Since mitophagy can be regulated also by mitofusin2, (Li J. Front Cell Dev Bio)  It is possible to validate mitofusin ubiquitination and parkin translocation on Mfn2.

Response 3: Thank you for pointing this out. We agree with this comment. Mfn1/2 was not used as a focus of the study as the computer analysis did not show it to be in the top five in terms of impact factor, but we will conduct future experiments related to this to verify that.

Comments 4: Since mitophagy is linked to the mitochondria fission/fusion process, the author needs to show mitochondria confocal pictures in LHON patients.

Response 4: Thank you for pointing this out. We agree with this comment. we will conduct future experiments related to this to verify that.

Comments 5: Discussion: Line 456 authors need to add more sentences related to mitochondria dynamism, there is new evidence on mitochondria fusion/ fission/ mitophagy and mitochondria clumping (Franco A. Life 2022)

Response 5: Thank you for pointing this out. We agree with this comment. As the focus of attention in this paper was centred on oxidative stress and mitophagy, experiments in mitochondrial dynamics were not covered and are not mentioned in the discussion.

Reviewer 3 Report

Comments and Suggestions for Authors

Xu et al used the immortalised skin fibroblast models derived from 2 LHON patients and 1 healthy control of unknown age/sex to perform the investigation on cell viability, evidence of oxidative stress and mitochondrial function, and the intervention of SkQ1.  The major drawback of this study is the limited number of representative lines.  Besides, generation of adult dermal fibroblast models does not require immortalisation or otherwise it may create abnormalities.  In additions, fibroblast models are exceptionally resistant to oxidative stress and that disfavours them as a cell model of neuronal tissues that are considered as vulnerable to oxidative stress.

Technically, albeit DCFH-DA is a popular method to detect ROS, it is no longer acceptable in the field owing to its artefactual results – DHE should have been used by a time-course ratiometric measurement.  H2O2 as a mean of introducing oxidative stress is a poor choice in comparison to the slower acting paraquat or rotenone.  Details are lacking in using mitoSOX to measure mitochondria superoxide – how did the authors use confocal to achieve a semi-quantitative analysis?  Again how did they quantify the JC-1 data?  They should also provide more details on their SeaHorse assay, such as cell counts – pre or post-assays, how?  The SeaHorse profile appears problematic considering the residual OCR of the LHON-II is lowered than the other 2 in fig1G.  The levels of toxins used were way too high for typical assays of this kind – should they describe the final conc of those toxins in the wells after injection?

In view of LHON is abnormal in mitochondrial DNA, should the authors examine the impact of SkQ1 treatment to mtDNA protection?

Minor points:

1) Intro: The SkQ1 conc in the mitochondrial matrix could be 1x108

2) M/M: I cannot find the growth conditions for the HEK and RPE-1 cells

Author Response

Comments 1: The major drawback of this study is the limited number of representative lines.  

Response 1: Thank you for pointing this out. We has demonstrated in previous studies that ROS are significantly elevated in skin cells of LHON compared to skin cells of healthy volunteers. Secondly, one of the main aims of this study was to demonstrate that SkQ1 could clear ROS in LHON cells and this was also verified in HEK293T cells. In addition, LHON is a rare disease, and the number of patients who enter our hospital for genetic testing and agree to skin collection is limited.

Comments 2: Generation of adult dermal fibroblast models does not require immortalisation or otherwise it may create abnormalities.

Response 2: Agree. Use of fibroblast cells is quite a common procedure in LHON studies [1]. However, the challenges of using biopsy-derived fibroblasts is that it take several weeks to culture sufficient cells for experimentation, and the cells have a limited proliferative lifespan [2]. To have access to a large resource of cells and to improve the predicament of low cell passaging and slow proliferation, we decided to construct immortalized fibroblasts, which would belong to the same population genetically and be a pool of cells with the same genetic traits. In addition, due to the differences in energy metabolism between fibroblasts and RGCs, we also used human retinal pigment epithelial cells (RPE-1) to verify.

[1] Jankauskaitė E, Bartnik E, Kodroń A. Investigating Leber's hereditary optic neuropathy: Cell models and future perspectives. Mitochondrion. 2017 Jan;32:19-26.

[2] Bahr T, Welburn K, Donnelly J, Bai Y. Emerging model systems and treatment approaches for Leber's hereditary optic neuropathy: Challenges and opportunities. Biochim Biophys Acta Mol Basis Dis. 2020 Jun 1;1866(6):165743.

Comments 3: Fibroblast models are exceptionally resistant to oxidative stress and that disfavours them as a cell model of neuronal tissues that are considered as vulnerable to oxidative stress.

Response 3: Agree. However, it's not permissible to obtain patients' retinal ganglion cells. Second, fibroblasts are thought to have more similarities with RGCs and can be cultured from skin biopsy without viral transformation. Like blood cells, an advantage of using fibroblasts is the representation of the nuclear genome of an LHON-afflicted patient. Cultured fibroblasts have been used to study protein expression, ATP production, and the effects of environmental factors in LHON, as well as respiration and ETC complex function [1].

[1] Bahr T, Welburn K, Donnelly J, Bai Y. Emerging model systems and treatment approaches for Leber's hereditary optic neuropathy: Challenges and opportunities. Biochim Biophys Acta Mol Basis Dis. 2020 Jun 1;1866(6):165743).

Comments 4: Technically, albeit DCFH-DA is a popular method to detect ROS, it is no longer acceptable in the field owing to its artefactual results – DHE should have been used by a time-course ratiometric measurement.

Response 4: Thank you for pointing this out. The DCFH-DA reactive oxygen species assay can be widely used in biomedical research, such as the study of cellular oxidative damage, inflammatory response, and antioxidant activity. In addition, DCFH-DA can be used to evaluate the antioxidant properties of drugs. This method has the advantages of being fast, sensitive and easy to use, and is widely used in cellular and molecular biology research fields [1].

In this study, DCF fluorescent images were acquired with a Nikon confocal microscopy equipped with 40×dry objective (excitation 488 nm, emission 495–550 nm) with fixed scanning parameters (Gain: 4×, exposure time: 50 ms). Unstained cells were background. The mean intensity of DCF fluorescence were analyzed by ImageJ software.

[1] Zhang Y, Li JJ, Xu R. Nogo-B mediates endothelial oxidative stress and inflammation to promote coronary atherosclerosis in pressure-overloaded mouse hearts. Redox Biol. 2023 Dec;68:102944.

Comments 5: H2O2 as a mean of introducing oxidative stress is a poor choice in comparison to the slower acting paraquat or rotenone.

Response 5: Thank you for pointing this out. Paraquat or rotenone have a competitive inhibitory effect with SkQ1, as confirmed by our early experiments. Some researchers have previously made the same point by conducting experiments [1].

[1] Antonenko YN, Avetisyan AV, Bakeeva LE. Mitochondria-targeted plastoquinone derivatives as tools to interrupt execution of the aging program. 1. Cationic plastoquinone derivatives: synthesis and in vitro studies. Biochemistry (Mosc). 2008 Dec;73(12):1273-87.

Comments 6: Details are lacking in using mitoSOX to measure mitochondria superoxide.

Response 6: Thank you for pointing this out. MitoSox (Invitrogen), a mitochondrion-specific hydroethidine-derivative fluorescent dye, was used to assess mitochondrial O2- production in cells in situ [1].

  1. Prepare a stock solution and working solution of MSR reagent. Make a 5 mM stock solution of MSR reagent by dissolving the contents of the vial in 13 µL of anhydrous DMSO. This stock solution is stable for one day.To make the working solution with the MSR reagent, add 5 µL of the 5 mM stock solution to 50 mL of HBSS with Calcium andMagnesium, or other buffer, to make 500 nM working solution.
  2. Apply 1–2 mL of the MSR reagent reagent working solution to cover cells adhering to coverslip(s) in a well of 35 mm dish.
  3. Incubate cells for 30 minutes at 37℃and 5% CO2. Protect from light.
  4. Wash cells gently 3 times with warm buffer (HBSS with Calcium and Magnesium or suitable buffer).
  5. View cells using spectral properties within 2 hours of staining.

[1] Ungvari, Zoltan et al. “Dysregulation of mitochondrial biogenesis in vascular endothelial and smooth muscle cells of aged rats.” American journal of physiology. Heart and circulatory physiology vol. 294,5 (2008): H2121-8.

Comments7: how did the authors use confocal to achieve a semi-quantitative analysis?

Response 7: Thank you for pointing this out. In this study, MitoSOX fluorescent images were acquired with a Nikon confocal microscopy equipped with 40×oil objective (excitation 396 nm, emission 610 nm) with fixed scanning parameters (Gain: 2×, exposure time: 120 ms). Unstained cells were background. The mean intensity of MitoSOX fluorescence were analyzed by ImageJ software.

Comments 8: Again how did they quantify the JC-1 data?  

Response 8: Thank you for pointing this out. We used imagej software to semi-quantitatively analyse the fluorescence images of JC-1. Imagej can measure the average fluorescence intensity of a specific region. The general steps are: open the image, split the channels, convert to 8 bit black and white image, set the threshold, and measure the average grey value. The formula for Mean Fluorescence Intensity is: Mean Fluorescence Intensity (Mean) = Sum of Fluorescence Intensity in the area (IntDen) / Area of the area (Area). 

The mitochondrial membrane potential was assessed using the ratio of the average fluorescence intensity of green fluorescence to that of red fluorescence, with a larger ratio indicating a lower mitochondrial membrane potential [1].

[1] Yao J, Liang X, Xu S, Liu Y, Shui L, Li S, Guo H, Xiao Z, Zhao Y, Zheng M. TRAF2 inhibits senescence in hepatocellular carcinoma cells via regulating the ROMO1/ NAD+/SIRT3/SOD2 axis. Free Radic Biol Med. 2024 Feb 1;211:47-62.

Comments 9: They should also provide more details on their SeaHorse assay, such as cell counts – pre or post-assays, how?  

Response 9: Thank you for pointing this out. 

(1) The day before the experiment

① Switch on and warm up: Switch on and warm up the Seahorse XFe96 detection system. Turn on the Seahorse XFe96 Controller by pressing the switch in the lower right corner of the Seahorse XFe96 Controller (control computer). Open the Wave software and wait until the controller is successfully connected to the instrument mainframe and warmed up to 37℃.

â‘¡ Add 20 mL of XF Calibrant to a 50 mL centrifuge tube and incubate overnight in a 37°C CO2-free cell incubator.

â‘¢ Hydration probe plate: remove the cover of the hydration plate and the probe plate and place them upside down on the lab bench (to protect the probes on the probe plate from damage). Add 200 μL of sterile water to each well of the hydration plate. Align the lid and the probe plate to each well of the hydration plate and restore the whole device by putting it back on the hydration plate.

â‘£ Inoculate cells: fibroblasts were spread on a special cell culture plate with approximately 1.5 × 104 cells per well and grown overnight at 37°C in a 5% CO2 incubator.

(2) On the first day of the experiment

① Replace the cell culture medium with XF medium, which contains 2 mM glutamine, 1 mM pyruvate and 10 mM glucose; and leave the cell culture plate at 37℃, CO2-free incubator for 1 h.

(ii) Oligomycin (15 µM), p-trifluoromethoxyphenylhydrazone (20 µM), and a mixture of antimycin A (5 µM) and rotenone (5 µM) were injected into the probe plates during the assay.

(iii) Normalisation of cell number.

Determination of protein concentration

â‘  Configure the protein concentration gradient with 0.5 mg/mL standard and add it into 96-well plate.

â‘¡ BCA working solution liquid preparation: BCA reagent A liquid and BCA reagent B liquid (50:1) mixed to configure the appropriate amount of BCA working solution, mix well;

â‘¢ Add 2 μL of each protein sample, add 18 μL of PBS dilution solution (10-fold dilution) to each well, and set up 3 replicate wells for each sample;

â‘£ Add 200 μL of BCA working solution to each well, and react for 30 min at 37℃ on a shaking table protected from light;

⑤ Select the wavelength of 562 nm for the enzyme labeller, and plot the standard curve according to the absorbance value of the detected standard protein solution. Based on the standard curve and the dilution times, the protein concentration of each sample to be detected can be calculated.

(iv) Analyses were performed using the software Wave 2.6.3.

Comments 10: The SeaHorse profile appears problematic considering the residual OCR of the LHON-II is lowered than the other 2 in fig1G.  The levels of toxins used were way too high for typical assays of this kind – should they describe the final conc of those toxins in the wells after injection?

Response 10: Thank you for pointing this out. Toxins concentration is fixed: Oligomycin (15 µM), p-trifluoromethoxyphenylhydrazone (20 µM), and a mixture of antimycin A (5 µM) and rotenone (5 µM) were injected into the probe plates during the assay.

Comments 11: In view of LHON is abnormal in mitochondrial DNA, should the authors examine the impact of SkQ1 treatment to mtDNA protection?

Response 11: Thank you for pointing this out.Previously, our team has explored this issue and has come to the following conclusions. This study explored whether multiple variations in mitochondrial genes were associated with the heterogeneity, mainly phenotypic heterogeneity. We found that cells (LHON-â…¡) with more mtDNA variants had higher ROS levels, lower mitochondrial membrane potential, and weaker respiratory function. Flow cytometry and cell viability testing showed that multiple mtDNA variants are associated with different levels of cell viability and apoptosis. In conclusion, we found that skin-derived fibroblast cells from G11778A LHON patients could be used as models for LHON research. Multi-mtDNA variants contribute to mitochondrial function variety, which may be associated with heterogeneity in patients with LHON[1].

[1] Yao S, Zhou Q, Yang M, Li Y, Jin X, Guo Q, Yang L, Qin F, Lei B. Multi-mtDNA Variants May Be a Factor Contributing to Mitochondrial Function Variety in the Skin-Derived Fibroblasts of Leber's Hereditary Optic Neuropathy Patients. Front Mol Neurosci. 2022 Jul 13;15:920221.)

Comments 12: Minor points:

1) Intro: The SkQ1 conc in the mitochondrial matrix could be 1x108

  • M/M: I cannot find the growth conditions for the HEK and RPE-1 cells

Response 1: Thank you for pointing this out.

  • The SkQ1 concentration in the mitochondrial matrix could be 1×108 times higher than in the extracellular space under certain conditions (Antonenko et al., 2009).
  • Added in the Methods section of the manuscript(Page 2, materials2.1 ):

HEK293T cells were maintained in DMEM high glucose medium (10-013-CV, Corning, NewYork, USA) containing 10% fetal bovine serum (35-030-CV, Corning, NewYork, USA) and 1% Penicillin-Streptomycin Solution 100× (P1400, Solarbio, Beijing, China). HTERT RPE-1 cells were cultured in DMEM (Dulbecco’s Modified Eagle’s Medium)/Hams F-12 50/50 Mix (10-092-CV, Corning, NewYork, USA) supplemented with 10% fetal bovine serum. The above three types of cells were cultured at 37°C in a 5% CO2 thermostatic incubator. HEK293T and HTERT RPE-1 cell lines were purchased from the American Type Culture Collection (ATCC, Manassas, VA, USA).